# Increasing Supply for Woody-Biomass-Based Energy through Wasted Resources: Insights from US Private Landowners

Quan-Hoang Vuong [1], Quang-Loc Nguyen [2], Ruining Jin [3,*], Minh-Hieu Thi Nguyen [4,5], Thi-Phuong Nguyen [6], Viet-Phuong La [1,7] and Minh-Hoang Nguyen [1]

1. Centre for Interdisciplinary Social Research, Phenikaa University, Yen Nghia Ward, Ha Dong District, Hanoi 100803, Vietnam; phuong.laviet@phenikaa-uni.edu.vn (V.-P.L.); hoang.nguyenminh@phenikaa-uni.edu.vn (M.-H.N.)
2. SP Jain School of Global Management, Lidcombe, NSW 2141, Australia
3. Civil, Commercial and Economic Law School, China University of Political Science and Law, Beijing 100088, China
4. School of Psychology, Massey University, Auckland 0745, New Zealand; hieuntm@hanu.edu.vn
5. Faculty of Management and Tourism, Hanoi University, Nam Tu Liem District, Hanoi 10000, Vietnam
6. Centre for Crop Systems Analysis, Wageningen University, P.O. Box 430, 6700 AK Wageningen, The Netherlands
7. A.I. for Social Data Lab (AISDL), Vuong & Associates, Hanoi 100000, Vietnam
* Correspondence: cu224004@cupl.edu.cn

**Abstract:** Woody biomass is suggested as a substitute for fossil fuels to achieve sustainable development. However, transitioning the land purpose to produce woody biomass entails investment and a tradeoff between wood pellet production and the current utilities created by the land, hindering the willingness of private landowners. To many forest landowners, forest trees and residues considered unprofitable to transport would be left in the forest without other proper use. The wasted woody resources on the land can be a potential source to increase the woody biomass supply. To support the policymakers, logging companies, state agencies, and landowners to better capitalize on these wasted resources, we aimed to identify the characteristics of woody-resource-wasting landowners and examine how to increase their likelihood to contribute to woody-biomass-based energy. By employing Bayesian Mindsponge Framework (BMF) analytics on a dataset of 707 private landowners in the United States (US), we discovered that landowners being male, having higher income, and being a member of a state/national forestry organization were more likely to waste woody resources. Moreover, woody-resource-wasting landowners perceiving woody-biomass-based energy as a substitution for fossil fuel were more likely to sell wood. In contrast, those perceiving environmental costs over the benefits of woody-biomass-based energy were less likely to sell. These findings can be used as insights for policymakers, logging companies, and state agencies to find an additional supply of woody-biomass-based energy from landowners likely to waste woody resources.

**Keywords:** woody biomass; environmental psychology; landowners; mindsponge theory; wasting behavior; environmental knowledge

## 1. Introduction

The Earth is approaching the tipping point, which can lead to large-scale discontinuities in the climate system when passed [1]. In a recent special report, the Intergovernmental Panel on Climate Change (IPCC) estimated that the tipping points could be passed if the global temperature exceeds 1.5 °C above pre-industrial levels [2]. Greenhouse Gas (GHG) emission is generally considered a contributor to climate change, and a considerable amount of GHG emission is generated during the combustion process of fossil fuels, where it releases carbon dioxide ($CO_2$) and other greenhouse gases into the atmosphere, trapping heat and leading to global warming [3]. To avoid the cascading effects of climate change,

the Paris Agreement requires governments to pledge to reduce Greenhouse Gas Emissions by 45% by 2030 and reach net zero by 2050. However, such an initiative faces tremendous resistance during policy implementation because lowered GHG emissions entail less use of fossil fuels—the energy source that remains the most prevalent in many countries [4]. When the usage of fossil fuels declines, new energy sources have to fill the void or resistance will occur due to socioeconomic backlash caused by power shortage. As a result, scientists and policymakers are seeking alternative energy sources that are largely available around the globe, able to offer social and economic interests, and are environmentally friendly [5,6]. Woody biomass is one of these alternatives.

According to Hodges et al. [7], woody biomass is a dispatchable and renewable energy source to generate heat. Some even believe it is currently one of the most promising alternatives to fossil fuels [8,9]. First of all, woody biomass is available in many places. It has been shown by extant research to have a high availability across different regions [10–13], lowering the difficulty of promoting woody biomass energy to replace fossil fuel energy. Secondly, woody biomass has positive social and economic impacts, as logging residues for bioenergy production can create jobs and income [14].

Woody-biomass-based energy also has environmental benefits [15–17]. According to the life cycle analysis (LCA) of GHG emissions of forest-derived electricity in the US, emissions from bioelectricity generation are significantly lower than the average emissions of existing electric systems (more than 95% lower, with 20.0 g $CO_2$e/kWh compared to 407.0 g $CO_2$e/kWh) [18]. Given the fact that the electric power sector in the US was the second contributor to GHG emissions (26.9%) in 2018 [19], and with the rapid growth of information technologies and electrification of transportation (e.g., electric vehicles) [20,21], woody-biomass-based energy will be more and more essential for meeting climate change prevention targets while keeping the economic and technological development trend. In addition, US landowners are advised to dispose of postharvest residues and waste wood (e.g., tops, limbs, and unmerchantable pulpwood) to reduce the risk of forest fires. Currently, the most common disposal method is on-site incineration [22,23]. However, this method is not only costly (∼\$370–\$2100/ha) [23] but also leads to air pollution, residue tree mortality, and reduction of site productivity for decades [22,24,25].

A typical example of woody biomass products is wood pellets, which are solid biofuels made from compressed cylindrical biomass particles [26]. Wood pellets partially replace oil or natural gas in many high-efficiency stoves and boilers for commercial, industrial, and domestic heating and power generation [27]. Due to the renewable energy regulations in the European Union (EU) and renewable portfolio standards in the United States (US) states, there has been a resurgence of interest in the potential role of wood-based bioenergy in promoting climate objectives and local economic development in the US [28–30]. Most of the wood pellets exported from the United States (US) to the United Kingdom, the Netherlands, and Belgium for power generation between 2009 and 2015 came from the southeastern US [31].

A majority of forest land is privately owned in the United States [29]; for these private landowners, cutting down trees for timber management/sale is the most common usage for their forest land [7,32]. A transition of land purposes from cutting down trees for timber to wood pellet production can be a way to enhance the production of wood pellets. Still, it entails investment and a tradeoff between wood pellet production and the current utilities created by the land [33]. Most landowners might be unwilling to make such a transition. Moreover, some criticize that the increase in wood pellet production will result in a decline in climate change and biodiversity loss [31].

One advantage of wood pellet production is that its supplying materials can be sourced from residuals for other purposes. Most wood pellets produced in the US are made from residual biomass from more established, higher-value forest products such as timber and paper; alternatively, wood pellets could also be made directly from wood in a forest [7]. According to the US Energy Information Administration, sawmill residuals and wood product manufacturing residuals accounted for approximately 70% of the wood pellet

feedstocks (more than 8 million tons) in 2022 [34]. Given that, to many forest landowners, forest trees and residues considered unprofitable to transport would be left in the forest without other proper use [31], those wasted woody resources on the land of the landowners can be a potential source to increase the supply for wood pellet production. Turning the left-in-forest woody waste into wood pellets might be a more attractive and feasible option for forest landowners because it does not require them to invest or make a tradeoff but might also help them to generate additional income.

Therefore, it is vital to discover characteristics of landowners with a high likelihood of wasting woody resources so that policymakers and woody-biomass producers can target potential suppliers and make strategies accordingly to incentivize participation in the woody biomass market among this population. Moreover, exploring the factors that can contribute to the increased likelihood of selling woody resources for biomass-based energy production among the wasting-woody-resources population is also helpful for (1) policymakers who implement efficient supply chain strategies, (2) logging companies that invest in specialized harvesting equipment, (3) organizations that work with private landowners, and (4) state agencies that generate functional future energy production strategies [33].

Although previous studies have indicated the value of woody biomass, such as availability [10–13], socioeconomic benefits [14], and environmental contribution of woody resource-made energy [18], many landowners are still wasting woody resources on their land [31]. Identifying factors that can influence the wasting-woody-resources landowners' decision to sell wood-based products for biomass energy is also necessary. Some studies have been conducted to find those factors among general landowner populations [33,35–37] but no studies have been conducted among wasting-woody-resources landowners.

Based on the reasons above, the current study had two main objectives:

1.　Identify characteristics of landowners that are likely to waste the woody resources on their lands.
2.　Identify factors influencing wasting-woody-resources landowners' likelihood to cut and/or remove trees for sale for woody-biomass-based energy.

For the second objective, we employed the Mindsponge Theory to construct models exploring factors influencing wasting-woody-resources landowners' likelihood to cut and/or remove trees for sale for woody-biomass-based energy [38]. Then, the constructed model was validated using Bayesian Mindsponge Framework (BMF) analytics on a dataset of 707 landowners in the southeastern US [39,40]. A description of abbreviations found in this study is shown in Table S1, available in the Supplementary Materials.

## 2. Methodology

### 2.1. Theoretical Foundation and Assumptions

Quan-Hoang Vuong and Nancy K. Napier coined the term "mindsponge" in a study on acculturation and global mindset [41]. The concept is viewed as a dynamic process or mechanism that reveals how a mind accepts new cultural values and discards old ones based on context. The authors constructed the conceptual model of the mindsponge mechanism using the metaphor of "the mind as a sponge that squeezes out inappropriate values and absorbs new ones that fit or complement to the context" [41]. The mechanism is built upon various theories and models, including self-affirmation theory [42], multi-filtering process [41], information processing model [43,44], trust [45,46], inductive attitude [47], acculturation model [48–50], etc. Recently, the mindsponge mechanism has been developed into Mindsponge Theory, a theory of information processing of minds that is constructed based on the newest evidence from brain and life sciences. The Mindsponge Theory's information-processing approach has been proven effective in explaining other psychological and behavioral issues [51–53].

According to the Mindsponge Theory, the mind is an information collection-cum-processor that collects and interacts with the surrounding infosphere. The information process of the mind consists of the following characteristics [38]:

- It reflects the natural patterns of systems in the biosphere.
- It is a dynamic process that is dynamically balanced.
- It involves cost–benefit evaluation, which aims to increase the perceived benefit and reduce the perceived cost of the system.
- It consumes energy and, thus, follows the principle of energy saving.
- It has goal(s) and priorities, depending on the demand of the system.
- Its fundamental purpose is to prolong the system's existence in one way or another, including survival, growth, and reproduction.

Physical structures are required to serve as platforms for information processing activities; in highly developed species, the brains provide this function. When information from the external environment is absorbed and integrated into the mind, it will be maintained in memory. There are two main types of information within a mind: highly trusted information and awaiting-evaluation information [54]. Highly trusted information is information that was previously evaluated by the mind and forms the mindset. During the filtering mechanism, subjective cost–benefit evaluation is conducted based on the content of the mind, mainly the mindset, to determine what information can enter or is ejected from the mind. If the perceived benefits outweigh the perceived costs, the value of the information is deemed positive and can be accepted and integrated into the mindset and vice versa.

The mind generates perceived values rather than being objectively and fundamentally related to the information that delivers them. When determining the value of information, the mind examines the possible influence of the information on other information, especially in terms of self-interest. In this scenario, the perceived value is the perceived effect. When a piece of information exists independently of other pieces of information, it has no meaning. To develop an associated meaning and decide the subjectively perceived value, a piece of information is constantly linked to and compared with other pieces of information [55].

Following this logic, we assumed that landowners' intention to cut/remove trees for sale for woody-biomass-based energy would be influenced by their perceptions of biomass energy's costs and benefits. Regarding environmental benefits, sustainable biomass production can contribute significantly to climate change mitigation by substituting fossil fuels and reducing the amount Greenhouse Gas emissions [17,30]. In addition, selling woody biomass can also create revenue but it requires the seller to know the existence of the market. Thus, if the landowner perceives woody-biomass-based energy as a potential substitute for fossil fuels, they are more likely to know the environmental and economic benefits of woody biomass. The perceived benefits influence their subsequent filtering system and make it easier to absorb information related to biomass production, resulting in a higher likelihood of cutting and/or removing trees for sale for woody-biomass-based energy.

Nevertheless, some landowners, environmental groups, and conservationists still think wood pellet production is detrimental to the environment. Their concerns are mainly about wood pellet production's effects on old-growth forests, bottomland forests, and climate change (primarily concerned with how management changes caused by wood-pellet production systems influence the forest carbon cycle) [56,57]. Thus, if landowners perceive that woody-biomass-based energy has more negative impacts on the environment than positive impacts, they are more likely to perceive woody biomass energy as costly and less likely to cut and/or remove trees for sale for woody-biomass-based energy.

*2.2. Model Construction*

2.2.1. Variable Selection and Rationale

The current study employed a dataset about forest landowners' perceptions of their land ownership and impact on forests [39]. The dataset was generated by a mail survey on the Atlantic Coastal Plain of the United States, which is the base of two major ports (Norfolk-Newport News—Virginia (NNV) and Savannah—Georgia (SAV) for the export of wood pellets to Europe. In general, 2972 private forest landowners who owned at least four hectares of forestland were contacted, and 707 participants completed the survey. The survey was designed to include forest characteristics, forest management situations,

knowledge and interests in woody biomass for energy production, and the respondents' socio-demographic nature.

The survey collection followed the random selection process. There were 1500 forest landowners randomly selected from 6000 qualified names in the NNV fuelshed and 1478 forest landowners randomly selected from 6000 qualified names in the SAV fuelshed. A modified Dillman approach was used to conduct the survey [58]. The survey was pretested with a small sample of Tennessee private forest landowners. A cover letter and revised questionnaire were mailed to 2978 forest owners after three modified questions' wording (1500 in the NNV fuelshed; 1478 in the SAV fuelshed). Due to the incomplete delivery of the surveys sent to 28 private forest landowners in the SAV fuelshed, they were excluded from the study population. A postcard urging respondents to complete the survey was sent ten days later to the entire survey population. A second letter and a duplicate of the questionnaire were mailed to everyone who had not responded to the initial survey three weeks after the postcard reminder. By the end of the survey cycle, 707 completed surveys were received, yielding a final response rate of 24.3%. The final dataset was peer-reviewed before being published in the open repository. The data descriptor is available to readers at https://www.sciencedirect.com/science/article/pii/S2352340919310297?via%3Dihub (accessed on 12 April 2023).

As reflected in Table 1, the variable *Age* is used to present the age of respondents by subtracting the current year from the year they were born. The *Sex* variable presents the biological sex of the respondent. The income of the forest landowner is indicated by the *Income* variable. *LandownerOrg* and *ForestryOrg* indicate whether the respondent is a member of the landowner association and forestry organization, respectively.

**Table 1.** Variable description.

| Variable | Description | Type of Variable | Value |
|---|---|---|---|
| *Age* | The age of the landowner | Numerical | NA |
| *Sex* | The biological sex of the landowner | Binary | 0: Female; 1: Male |
| *LandownerOrg* | Whether the landowner participates in state/national landowner association | Binary | 0: No; 1: Yes |
| *ForestryOrg* | Whether the landowner participates in state/national forestry organization | Binary | 0: No; 1: Yes |
| *Income* | The income of the landowner's household before taxes in 2015 | Numerical | 1: Less than \$30,000; 2: \$30,000 to less than \$60,000; 3: \$60,000 to less than \$90,000; 4: \$90,000 to less than \$120,000; 5: \$120,000 to less than \$150,000; 6: \$150,000 or more |
| *WastedResources* | Whether the respondent is a woody-resource-wasting landowner | Binary | 0: No; 1: Yes |
| *AlternativetoFossil* | The landowner's level of agreement/disagreement with the statement: 'Woody-biomass-based energy is a viable alternative to fossil fuels' | Numerical | 1: strongly disagree; 2: somewhat disagree; 3: neither agree nor disagree; 4: somewhat agree; 5: strongly agree |

**Table 1.** *Cont.*

| Variable | Description | Type of Variable | Value |
|---|---|---|---|
| *CostsOverBenefits* | The landowner's level of agreement/disagreement with the statement: 'Woody-biomass-based energy has more environmental costs than benefits' | Numerical | 1: strongly disagree; 2: somewhat disagree; 3: neither agree nor disagree; 4: somewhat agree; 5: strongly agree |
| *FutureSaleforBiomass* | The landowner's likelihood of cutting and/or removing trees for sale for woody-biomass-based energy in the next five years | Numerical | 1: very unlikely; 2: somewhat unlikely; 3: neutral; 4: somewhat likely; 5: very likely |

The variable *AlternativetoFossil* indicates the forest landowners' perceptions toward woody-biomass-based energy being a viable alternative to fossil fuels, while the variable *CostsOverBenefits* represents the perception toward woody-biomass-based energy having more environmental costs than benefits. The variables were measured by a scale ranging from 1 to 5, with 1 being 'strongly disagree', and 5 indicating 'strongly agree.'

The *WastedResources* variable reflects whether the landowner wasted woody resources during the past five years. The variable was generated by asking the following question: 'If trees were cut and/or removed for any reason during the past five years, what was done with the harvest residuals (limbs, tops, etc.)?' Respondents who chose any of the following options were considered woody-resource-wasting landowners: 'Left in the woods where cut', 'Left at the landing area where logs were loaded onto trucks', 'Piled and burned.'

The respondent's likelihood of cutting and/or removing trees for sale for woody-biomass-based energy is represented by the *FutureSaleforBiomass* variable, with the scale ranging from 1 ('very unlikely') to 5 ('very likely').

### 2.2.2. Statistical Models

For fulfilling the first research objective, we formulate Model 1 with *WastedResources* being the outcome variable and the socio-demographic features of landowners as predictor variables. To help policymakers, logging companies, and state agencies capitalize on the wasted woody resources, we purposefully selected features that can increase the feasibility of later identification of landowners, such as age, sex, income, and being a member of associations. Model 1 was constructed as follows:

$$WastedResources \sim normal(\mu, \sigma) \tag{1a}$$

$$\mu_i = \beta_0 + \beta_{Age} * Age_i + \beta_{Sex} * Sex_i + \beta_{Income} * Income_i + \beta_{LandownerOrg} \\ * LandownerOrg_i + \beta_{ForestryOrg} * ForestryOrg_i \tag{1b}$$

$$\beta \sim normal(M, S) \tag{1c}$$

The logical network of Model 1 is displayed in Figure 1.

The probability around $\mu$ is determined by the form of the normal distribution, whose width is specified by the standard deviation $\sigma$. $\mu_i$ indicates the likelihood of landowner $i$ cutting and/or removing trees for sale for woody-biomass-based energy in the next five years; $Age_i$ indicates landowner $i$'s age; $Sex_i$ indicates landowner $i$'s biological sex; $Income_i$ indicates landowner $i$'s income; $LandownerOrg_i$ indicates whether landowner $i$ is a member of a state/national landowner association; $ForestryOrg_i$ indicates whether landowner $i$ is a member of a state/national forestry association. Model 1 has seven parameters: the coefficients, $\beta_{Age}$, $\beta_{Sex}$, $\beta_{Income}$, $\beta_{LandownerOrg}$, and $\beta_{ForestryOrg}$; the intercept, $\beta_0$; and the standard deviation of the "noise", $\sigma$. The coefficients of the predictor variables

are distributed as a normal distribution around the mean denoted *M* and with the standard deviation denoted *S*.

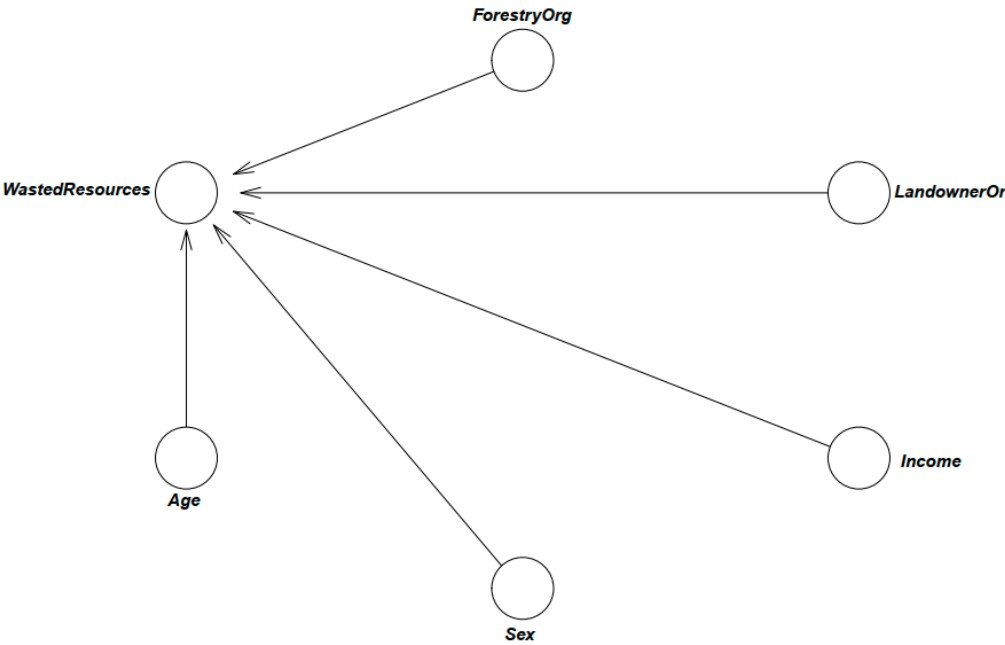

**Figure 1.** Logical network of Model 1.

To test the assumptions proposed in Section 2.1, we constructed Model 2. Model 2 was tested on the dataset of wasting-woody-resources landowners, which was extracted from the full dataset. In total, the dataset of wasting-woody-resources landowners consists of 361 landowners who chose the following options when being asked 'If trees were cut and/or removed for any reason during the past five years, what was done with the harvest residuals (limbs, tops, etc.)?': 'Left in the woods where cut', 'Left at the landing area where logs were loaded onto trucks', 'Piled and burned.'

$$FutureSaleforBiomass \sim normal(\mu, \sigma) \tag{2a}$$

$$\mu_i = \beta_0 + \beta_{AlternativetoFossil} * AlternativetoFossil_i + \beta_{CostsOverBenefits} \\ * CostsOverBenefits_i \tag{2b}$$

$$\beta \sim normal(M, S) \tag{2c}$$

$AlternativetoFossil_i$ indicates landowner *i*'s perception of woody-biomass-based energy as an alternative to fossil fuel; $CostsOverBenefits_i$ indicates landowner *i*'s perception of woody-biomass-based energy's environmental cost. Model 2 has four parameters: the coefficients, $\beta_{AlternativetoFossil}$ and $\beta_{CostsOverBenefits}$; the intercept, $\beta_0$; and the standard deviation of the "noise", $\sigma$. If the coefficient of $AlternativetoFossil$ is positive and the coefficient of $CostsOverBenefits$ is negative, our assumptions based on the mindsponge reasoning would be validated. Figure 2 demonstrates the logical network of Model 2.

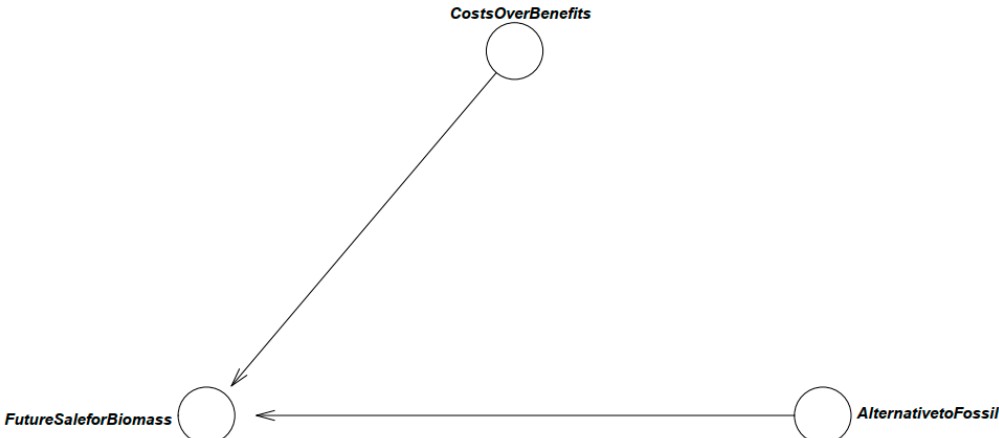

**Figure 2.** Logical network of Model 2.

### 2.3. Analysis and Validation

Bayesian Mindsponge Framework (BMF) analytics is an innovative analytical approach used in cognitive and psychological research. It integrates the Mindsponge Theory, which accounts for the dynamics and multiplexity of human thoughts and behaviors, with Bayesian analysis, which provides flexibility in model construction and fitting [40]. The Mindsponge Theory focuses on the underlying level of human psychology and behavior through the information-processing lens but not the high levels (observations at individual and societal levels); so, it does not contradict existing psychological and social theories and frameworks but rather helps elaborate, solve inconsistencies, and connect concepts through the dynamic view of information processing. Specifically, the Mindsponge Theory's reasoning has been effectively used to examine and explain psychological processes and mechanisms in various disciplines, such as mental health, psychological adaptation, and especially environmental psychology [51,53,59,60].

The framework was employed in the current study for several reasons. First, the approach combines the mindsponge theory's strengths in reasoning with Bayesian analysis' inference advantages [40]. Second, Bayesian inference considers all properties probabilistically (including unknown parameters) [61,62], enabling accurate prediction using parsimonious models. Bayesian approaches can still be used to fit a variety of complex models, including the polynomial model and non-linear regression structures, thanks to the advantage of the Markov chain Monte Carlo (MCMC) technique [63]. Third, compared to the frequentist approach, Bayesian inference has some advantages. For instance, it enables users to interpret results using credible intervals and the highest probability of parameters rather than the dichotomous decision of rejecting and accepting a hypothesis based on the *p*-value [64].

The research procedure is shown in Figure 3. Models with uninformative priors or a flat prior distribution were built to provide as little prior information for model estimations due to the exploratory nature of this study [65]. We used Pareto-smoothed importance sampling leave-one-out (PSIS-LOO) diagnostics to test the models' goodness-of-fit with the data after they had been constructed [66,67]. The mathematical expression of LOO is presented in Equation (3):

$$LOO = -2LPPD_{loo} = -2s \sum_{i=1}^{n} \log \int p(y_i|\theta) p_{post(-i)}(\theta) d\theta \qquad (3)$$

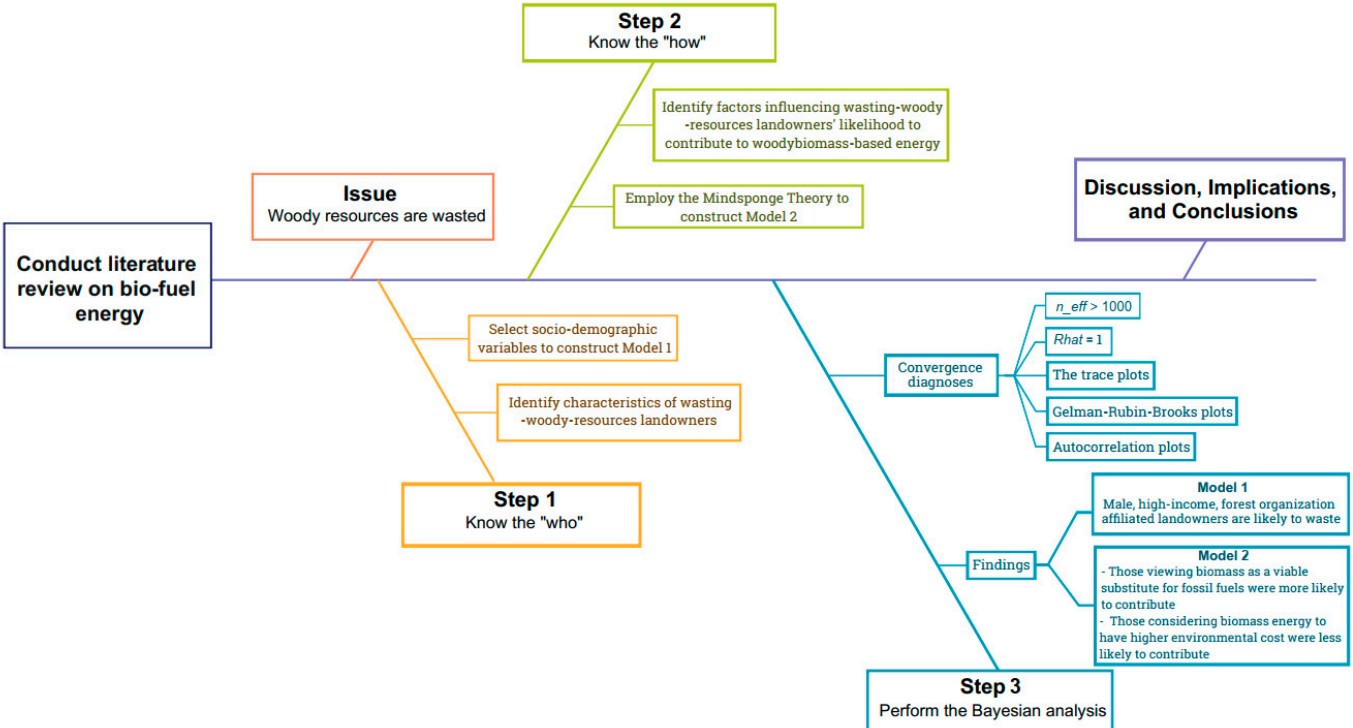

**Figure 3.** Flowchart of the research procedure.

The posterior distribution derived from the data minus the data point *i* is $p_{post(-i)}(\theta)$. The PSIS method for computing leave-one-out cross-validation in the R **loo** package uses *k*-Pareto values to help identify observations that have a significant impact on the PSIS estimate. The leave-one-out cross-validation may be tricky to compute accurately for observations with *k*-Pareto values greater than 0.7, which are frequently regarded as influential. A commonly accepted measure to consider a model having goodness-of-fit is that *k* values are below a threshold of 0.5.

The convergence of Markov chains can then be verified statistically using the effective sample size (*n_eff*) and the Gelman–Rubin shrink factor (*Rhat*), and graphically using trace plots, Gelman–Rubin–Brooks plots, and autocorrelation plots. The number of iterative samples that are not autocorrelated during stochastic simulation is represented by the *n_eff* value. The Markov chains are generally considered to be convergent if *n_eff* is greater than 1000, at which the effective sample size is adequate for inference [68]. The *Rhat* value, also known as the potential scale reduction factor or the Gelman–Rubin shrink factor, must not be higher than 1.1 for the model to converge [69]. If *Rhat* is equal to 1, the model can be deemed convergent.

The **bayesvl** R package is employed to perform Bayesian analysis. The package also produces high-quality visualization graphics and user-friendly operations [54]. All the code and data used for this study's analysis have been deposited on the Open Science Framework for public review and evaluation for transparency and cost-effectiveness [70] https://osf.io/kahwn/ (accessed on 12 April 2023).

## 3. Results

### 3.1. Model 1: Characteristics of Woody-Resource-Wasting Landowners

In the first model, we aimed to examine the effects of the landowners' age, gender, income, and participation in a state/national landowner association and forestry organizations on their likelihood to waste woody resources on their land. The PSIS diagnostic plot shows that all *k* values are below 0.25, suggesting that Model 1 has a high goodness-of-fit with the current data (see Figure 4).

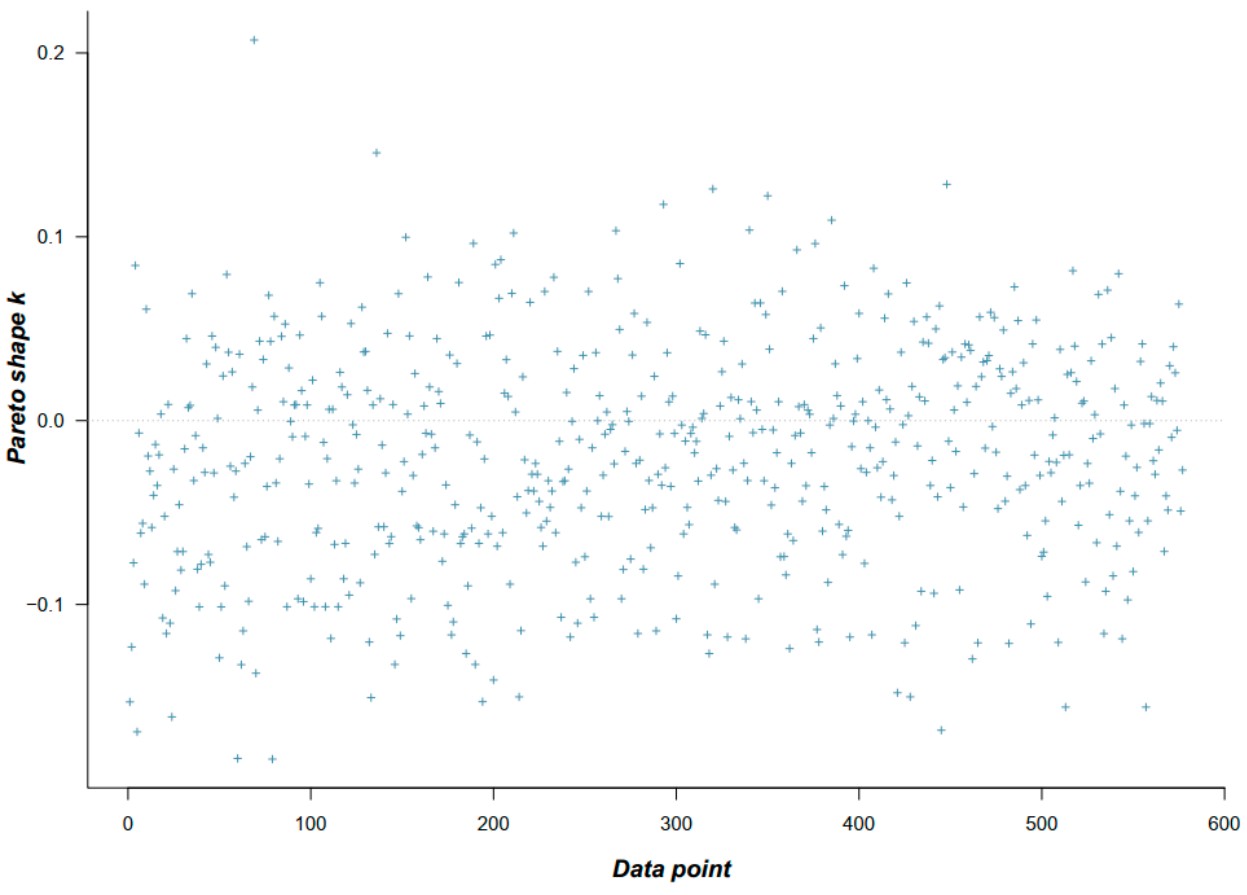

**Figure 4.** Model 1's PSIS diagnostic plot.

The effective sample size (*n_eff* > 1000) and Gelman–Rubin shrink factor (*Rhat* = 1) of all simulated posterior coefficients indicate a good convergence of the model's Markov chains (see Table 2). The trace plots of Model 1 are shown in Figure 5. The colored lines in the middle of trace plots are Markov chains. The Markov chains fluctuate around a central equilibrium after the warmup period (2000 iterations); so, they are deemed good-mixing and stationary. These two characteristics convey a good signal of Markov chain convergence.

**Table 2.** Model 1's simulated posteriors.

| Parameters | Mean (M) | Standard Deviation (S) | *n_eff* | *Rhat* |
| :---: | :---: | :---: | :---: | :---: |
| *Constant* | −1.96 | 0.48 | 7852 | 1 |
| *Age* | 0.01 | 0.01 | 10,642 | 1 |
| *Sex* | 0.36 | 0.20 | 11,533 | 1 |
| *Income* | 0.08 | 0.05 | 9862 | 1 |
| *LandownerOrg* | −0.02 | 0.27 | 9354 | 1 |
| *ForestryOrg* | 1.01 | 0.27 | 8911 | 1 |

The Gelman–Rubin–Brooks plots in Figure 6 show good convergence of the Markov chains through the rapid decline of the shrink factors to 1. Another visual convergence diagnosis, using the autocorrelation plots, also confirms the Markov chains convergence by illustrating the reduction of the autocorrelation levels to 0 among iterative samples (see Figure 7).

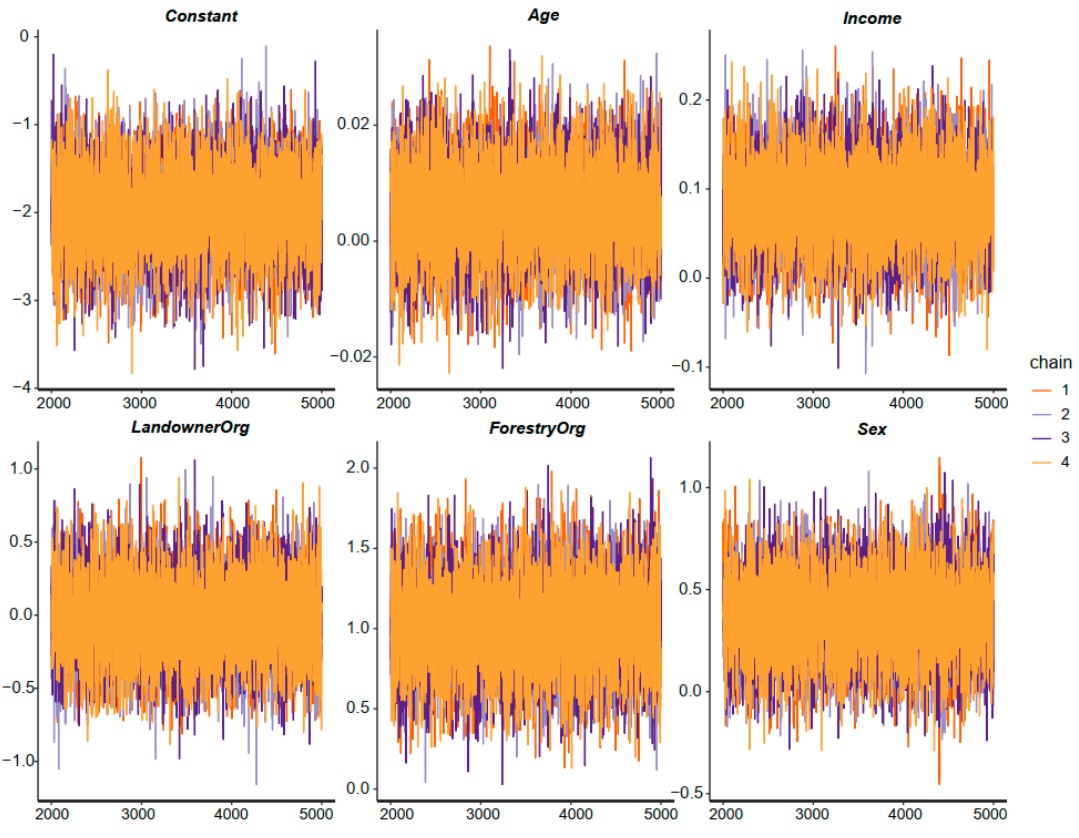

**Figure 5.** Model 1's trace plots.

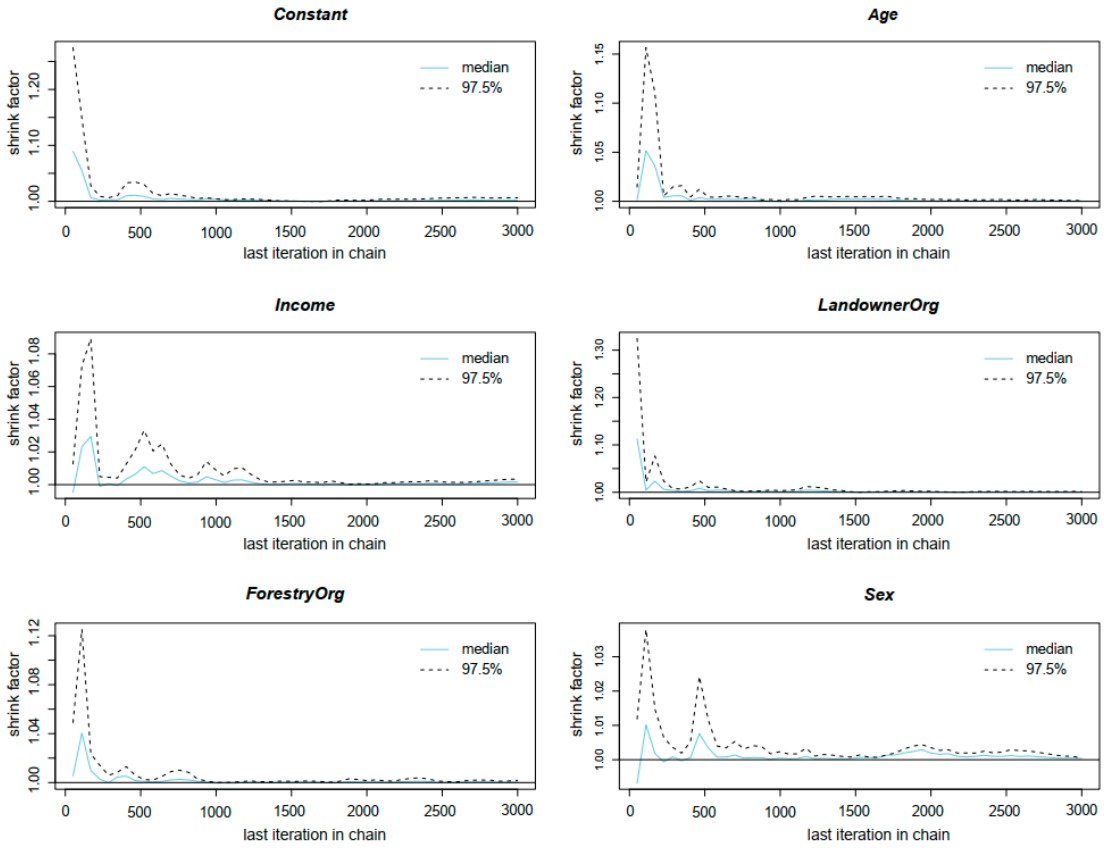

**Figure 6.** Model 1's Gelman–Rubin–Brooks plots.

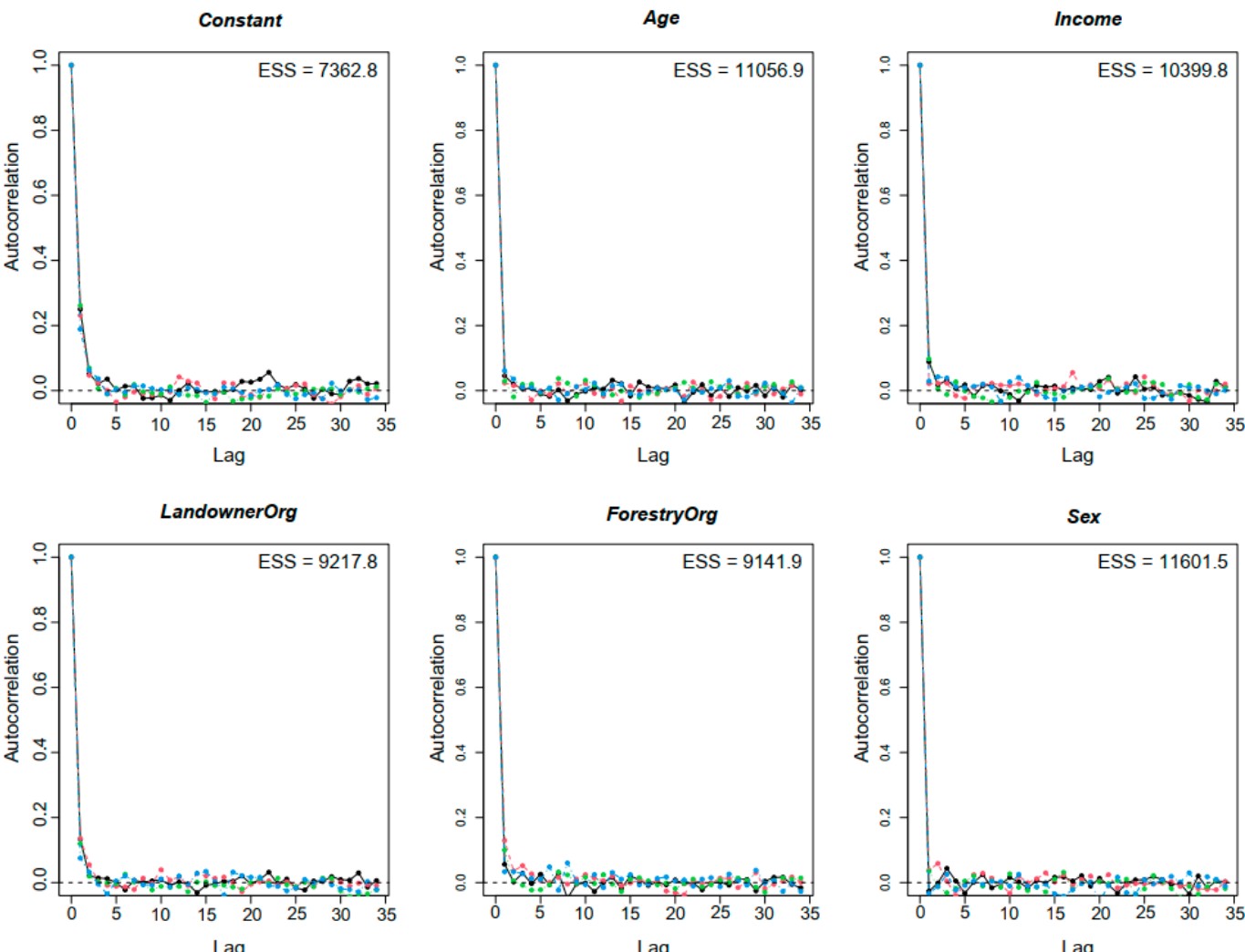

**Figure 7.** Model 1's autocorrelation models.

Figure 8 shows the posterior distributions of Model 1 and their Highest Posterior Density Intervals (HPDI) at 89%, represented by the thick black line in the middle of the histogram. The HPDI shows the credible range encompassing 89% of the estimates; so, the coefficient value within this range has the highest probability of happening. From the simulated posterior results of Model 1, we found that the older landowners were more likely to waste wood on the lands ($M_{Age}$ = 0.01 and $M_{Age}$ = 0.01). However, the effect is weakly reliable as a large proportion of its posterior distribution is still located on the negative side. In addition, the landowners having a higher income and being male were more likely to waste woody resources ($M_{Income}$ = 0.08 and $M_{Income}$ = 0.05; $M_{Sex}$ = 0.36 and $M_{Sex}$ = 0.20). The effects of income and sex are highly reliable as their HPDIs are located entirely on the positive side. Being a member of national/state organizations of forestry was also positively associated with the landowners' likelihood to waste the woods ($M_{ForestryOrg}$ = 1.01 and $M_{ForestryOrg}$ = 0.27); however, being a member of national/state organizations of landowners did not have any effect ($M_{LandownerOrg}$ = −0.02 and $M_{LandownerOrg}$ = 0.27).

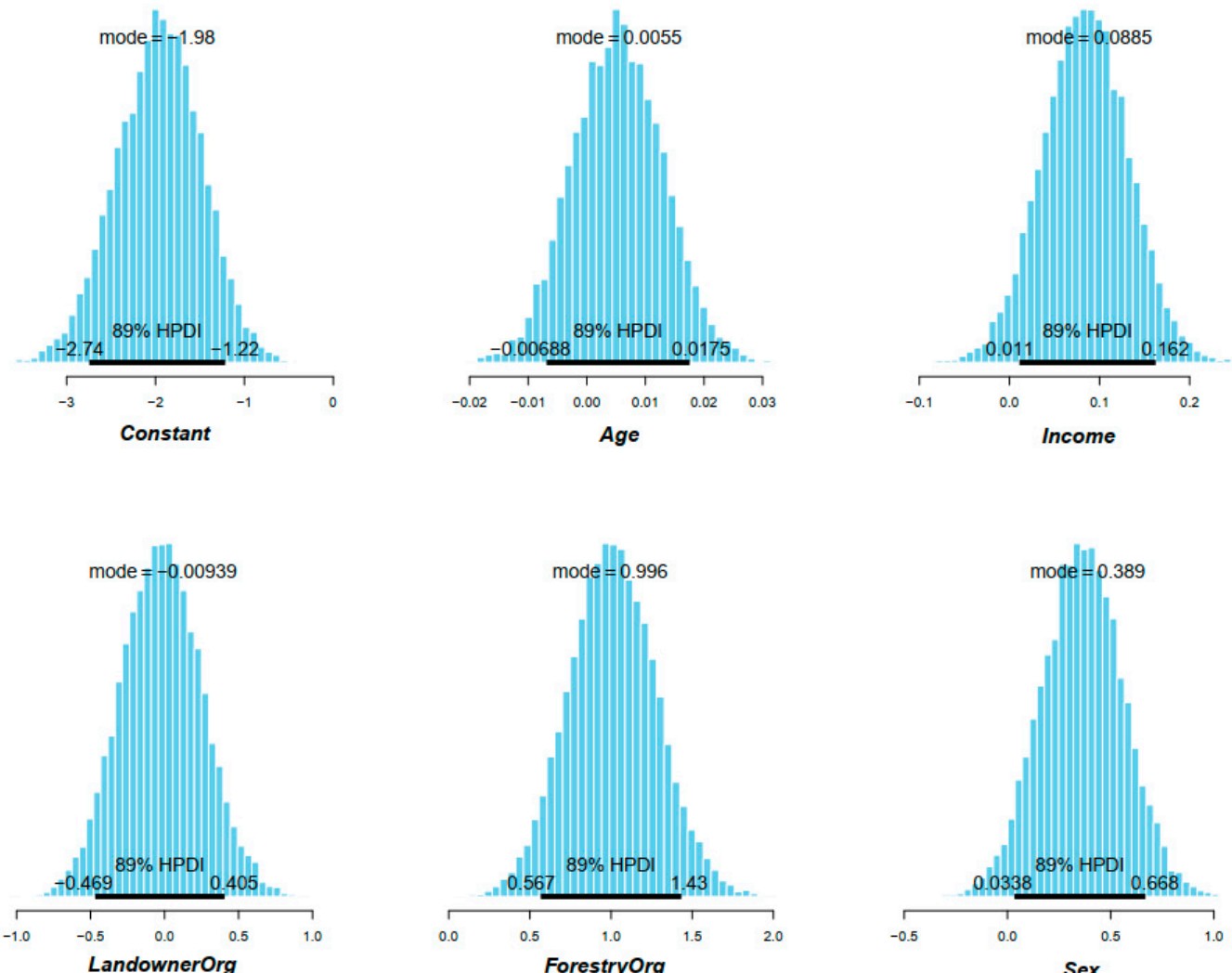

**Figure 8.** Model 1's posterior distributions.

*3.2. Model 2: Psychological Factors Influencing Woody-Resource-Wasting Landowners' Likelihood to Contribute to Woody-Biomass-Based Energy*

In the second model, we aimed to examine the effects of the landowners' perceptions of woody-biomass-based energy's cost and benefit on their likelihood to sell wood for woody-biomass-based energy in the next five years. The constructed model fits well with the dataset, as indicated by the PSIS diagnostic plot (all *k*-values < 0.15, see Figure 9).

The effective sample size (*n_eff* > 1000) and Gelman–Rubin shrink factor (*Rhat* = 1) of all simulated posterior coefficients indicate a good convergence of Model 2's Markov chains (see Table 3). The graphical convergence diagnostics, such as the trace plots (see Figure S1), Gelman–Rubin–Brooks plots (see Figure S2), and autocorrelation plots (see Figure S3), also confirm the convergence.

**Table 3.** Model 2's simulated posteriors.

| Parameters | Mean (M) | Standard Deviation (S) | *n_eff* | *Rhat* |
|:---:|:---:|:---:|:---:|:---:|
| *Constant* | 1.19 | 0.48 | 4975 | 1 |
| *AlternativetoFossil* | 0.47 | 0.09 | 5162 | 1 |
| *CostsOverBenefits* | −0.10 | 0.09 | 7497 | 1 |

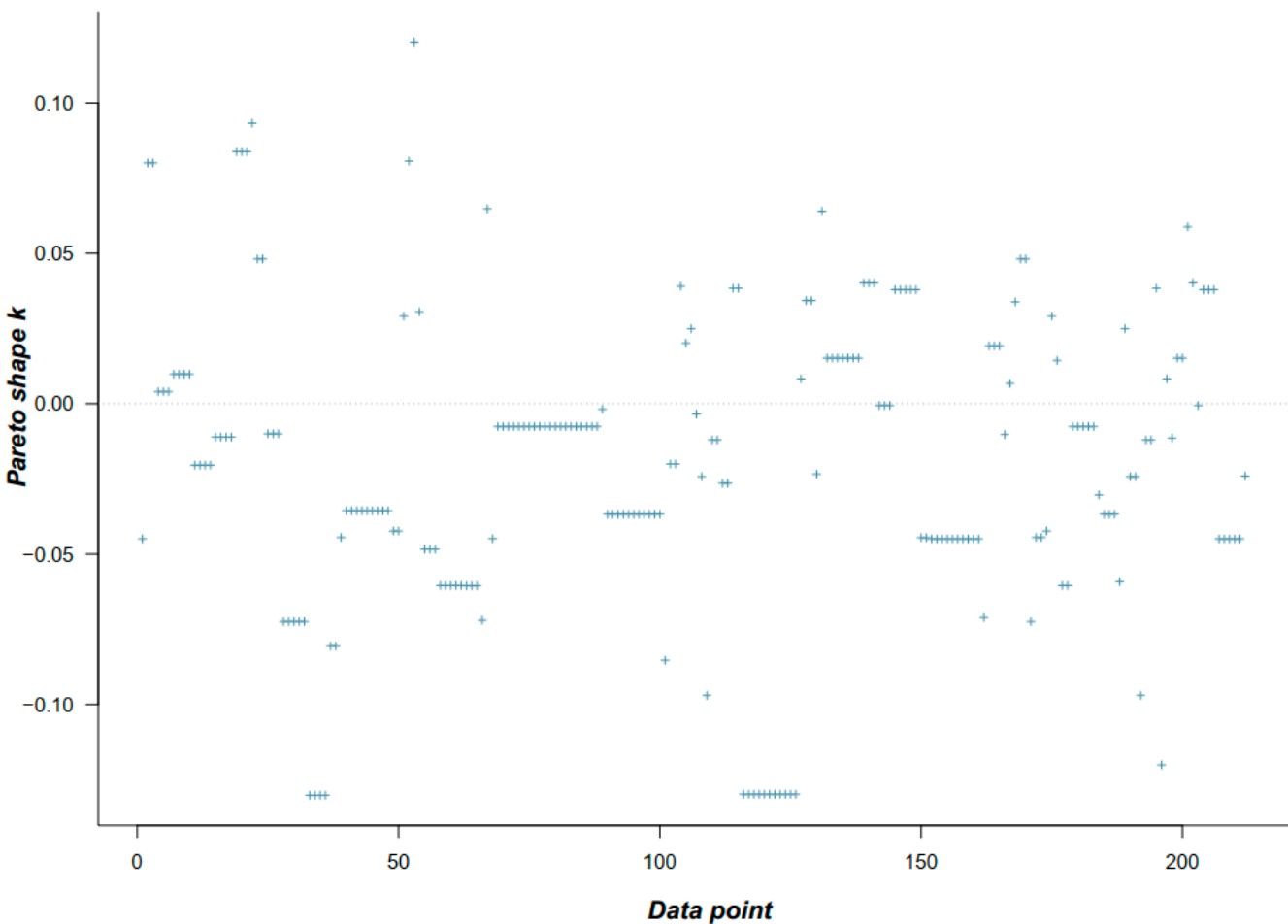

**Figure 9.** Model 2's PSIS diagnostic plot.

The posterior distributions of Model 2 are shown in Figure 10. The thick blue line represents the distribution's HPDI at 89%, while the thin line represents the remaining distribution. The dot in the middle of the distributions indicates the mean value, where the coefficient has the highest probability of occurring. Based on the estimated results, we found that landowners perceiving woody-biomass-based energy as a replacement for fossil fuel were more likely to sell their wood for woody-biomass-based energy ($M_{AlternativetoFossil} = 0.47$ and $M_{AlternativetoFossil} = 0.09$). The effect is highly reliable (see the distribution in Figure 10). In contrast, landowners perceiving the environmental costs over the benefits of woody-biomass-based energy were less likely to sell their wood for biomass-based energy ($M_{CostsOverBenefits} = -0.10$ and $M_{CostsOverBenefits} = 0.09$). Although a portion of *CostsOverBenefits*'s distribution lies on the positive side, that portion is small, so the effect can still be considered moderately reliable.

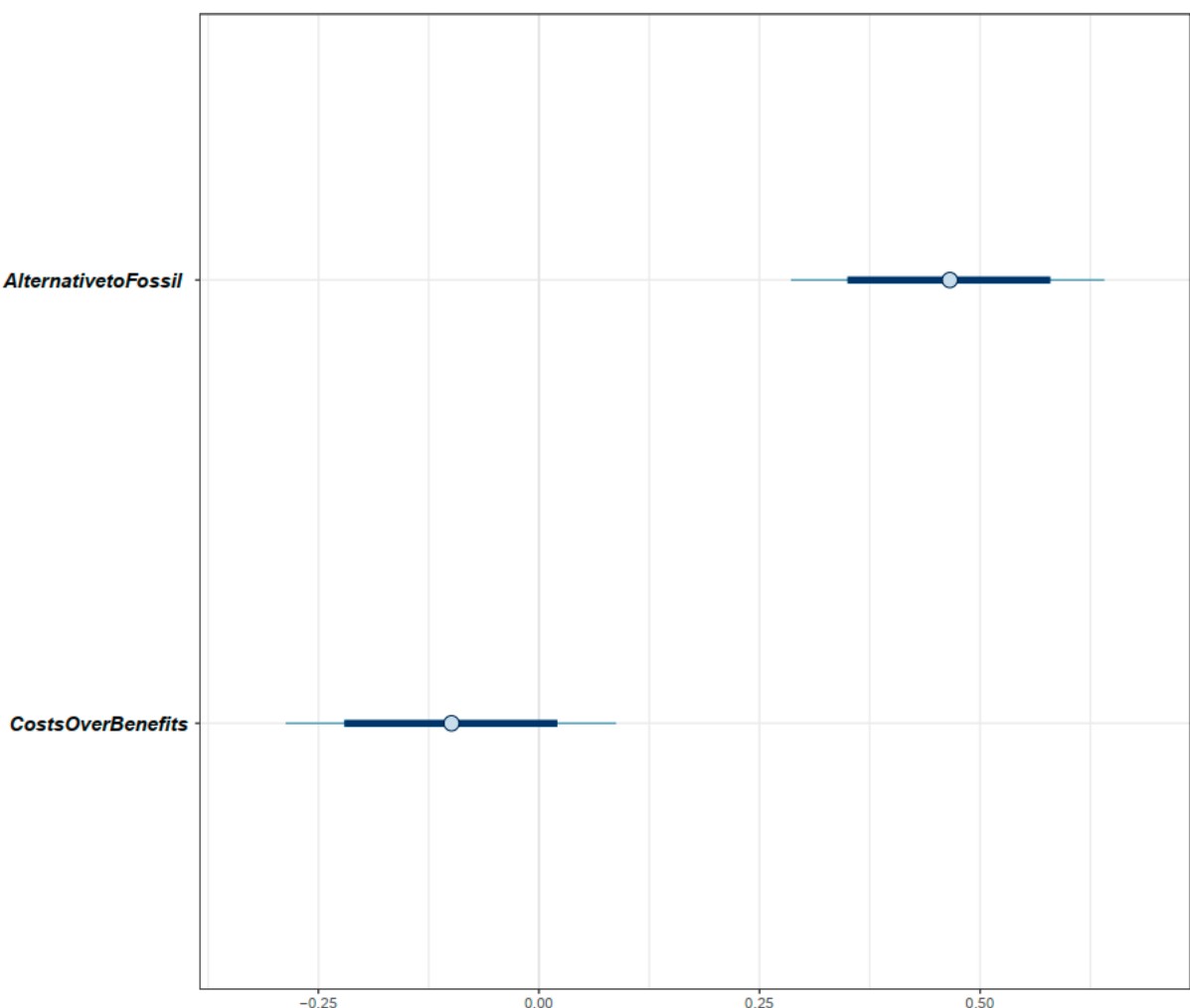

**Figure 10.** Model 2's posterior distributions.

## 4. Discussion

The current study employed BMF analytics on a dataset of 707 landowners in the southeastern US to discover (1) the characteristics of landowners likely to waste woody resources and (2) the psychological factors influencing wasting-woody-resources landowners' likelihood to cut and/or remove trees for sale for woody-biomass-based energy in the next five years. We found that landowners being male, having higher income, and being a member of a state/national forestry organization were more likely to waste woody resources. Although age also positively affected the wasting likelihood, the effect was weakly reliable.

The positive association between income and woody-resource-wasting likelihood could come from the landowners' subjective cost–benefit judgment. More specifically, most forest landowners value the natural beauty, privacy, and other benefits of the forests; therefore, timber-related production is not their main objective [71,72]. Even for the current study's sample size, the three most important reasons the landowners maintained their land as forest were to protect nature for wildlife habitat, to enjoy beauty or scenery, and to pass land on to their children or other heirs. Almost 70% of landowners (489 responses) reported that beauty or scenery enjoyment was an important or very important reason. In comparison, more than 71% of landowners (504 responses) thought natural protection for wildlife habitats was the reason. Timber production was only ranked fourth on the list of reasons [7]. It is plausible to say wealthy landowners are less likely to harvest and utilize timber and produce woody biomass energy when financial gains are not their primary

motivations [73,74], thus wasting woody resources on their forest land. The effects of sex and participation in a state/national forestry organization on wasting behaviors might need further studies for the explanation, as the sex difference was associated with many psychological differences, and forestry organization participation was associated with many contextual and social factors. For now, they can be used to identify landowners likely to waste woody resources on their lands.

Besides identifying the characteristics of landowners with a high likelihood to waste woody resources, we also discovered that cost–benefit perceptions associated with woody-biomass-based energy could influence woody-resource-wasting landowners' likelihood to sell wood for woody-biomass-based energy in the next five years. Specifically, landowners perceiving woody-biomass-based energy as a substitution for fossil fuel were more likely to sell wood, while those perceiving environmental costs over benefits of woody-biomass-based energy were less likely to sell. These results are consistent with the previous study of Sun et al. [75], who found a correlation between the forestland owner's understanding of the benefits of biomass and their decision to supply wood-based feedstocks for bioenergy. Similarly, a survey by Shaw et al. [76] on 395 forest landowners in North Carolina also showed that an education program for landowners with a low level of knowledge on woody biomass would lead to more positive attitudes and motivation to harvest woody biomass.

Drawing from the mindsponge information-processing approach, individuals interpret and process feedback from the interaction with the external environment through the lens of their subjective perceptions [38,55]. Information that aligns with their core values and beliefs (or highly trusted values) is synthesized and integrated; otherwise, they will carefully evaluate its costs and benefits before accepting or rejecting it. Most landowners in our study cared most about the environment as they maintained the land as a forest for enjoying nature and conserving biodiversity [7]. These features are the expression of the eco-surplus culture [77]: "a set of pro-environmental attitudes, values, beliefs, and behaviors that are shared by a group of people to reduce negative anthropogenic impacts on environments as well as conserve and restore nature" [52]. Influenced by the eco-surplus culture, their value systems prioritized the environment's health over other aspects. However, the effects of cost–benefit perceptions associated with woody-biomass-based energy might differ for landowners whose value systems are profit-driven.

## 5. Conclusions and Recommendations

The key objective of this exploration was to unearth the characteristics associated with landowners who frequently waste woody resources, with a particular focus on enhancing their propensity to contribute to woody-biomass-based energy in the future. This research study employed Bayesian Mindsponge Framework (BMF) analytics to analyze a dataset of 707 US private landowners. The results indicate that landowners who were male, had higher incomes, and were active members of state or national forestry organizations were significantly more prone to waste woody resources. Moreover, the study also indicates that landowners who viewed woody-biomass-based energy as a viable and feasible alternative to fossil fuel were more inclined to sell woody resources. Conversely, landowners who perceived the environmental costs of producing energy from woody biomass to overshadow its benefits exhibited a decreased propensity to sell.

Based on these findings, we suggest policymakers, logging companies, and state agencies find an additional supply of woody-biomass-based energy from landowners likely to waste woody resources. Those landowners are more likely to be males, high-income people, and state/national forestry organization members. Improving woody-resource-wasting landowners' knowledge of woody-biomass-based energy is essential to enhance their probability of supplying resources for woody-biomass-based energy production. Moreover, as the landowners tend to be people with eco-surplus culture, persuading them that woody-biomass-based energy production might not negatively affect the environment is imperative. To do so, more studies on the effects of woody-biomass-based energy on the environments, such as that of Dale, Parish, Kline, and Tobin [31], are required.

## 6. Limitations

The current study is not without limitations [78]. First of all, although our results indicate being a member of a state/national forestry organization is a signal of woody-resource-wasting landowners, we could not provide further details of those organizations. This could be a potential research direction in the future. Second, as we only fitted Model 2 using the dataset of 361 woody-resource-wasting landowners, the sample size is acceptable but not representative. Thus, results estimated using Model 2 should be validated with a larger sample size.

**Supplementary Materials:** The following supporting information can be downloaded at: https://www.mdpi.com/article/10.3390/su15118667/s1, Table S1. Table of abbreviations; Figure S1: Model 2's trace plots; Figure S2: Model 2's Gelman-Rubin-Brooks plots; Figure S3: Model 2's autocorrelation plots.

**Author Contributions:** Conceptualization, Q.-H.V. and M.-H.N.; data curation, R.J., M.-H.T.N. and T.-P.N.; formal analysis, Q.-H.V. and Q.-L.N.; methodology, M.-H.T.N. and V.-P.L.; supervision, M.-H.N.; writing—original draft, Q.-H.V. and M.-H.N.; writing—review and editing, Q.-L.N. and R.J. All authors have read and agreed to the published version of the manuscript.

**Funding:** This research received no external funding.

**Informed Consent Statement:** The participants were informed about the ethical standards, issues of information nondisclosure, and the possible insights the survey may contribute to the understanding of policymakers and the public in general. The participants had to provide informed consent before responding to the interviews.

**Data Availability Statement:** All the code and data used for this study's analysis have been deposited on the Open Science Framework for public review and evaluation for transparency and cost-effectiveness: https://osf.io/kahwn/ (accessed on 1 April 2023).

**Conflicts of Interest:** The authors declare no conflict of interest.

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
