# Peer review of "Increasing Supply for Woody-Biomass-Based Energy through Wasted Resources: Insights from US Private Landowners"

_sustainability, doi:10.3390/su15118667_

Round 1
Reviewer 1 Report
The paper was revised according to the journal rules. The topic treated was focused on the wood biomass considered as an energy source.
Few revisions are required and they are reported below:
- please try to reduce acronyms from the abstract
- I suggest to revise the keywords
-a nomenclature list with all acronyms and parameters should be added
- please try to link better the first part of the introduction section with the present study
- highlight how to reduce CO2 emission with woody biomass, an LCA approach should be considered - please add more information
- assumption should be clarified better in section 1
- I suggest to improve the discussion section considering the previous data and adding the range of data shown
- please add the number of equations
- figures 5 and 6 should be revised
- figures add in the suppl. material should be revised
The quality is moderately high
Author Response
Dear Sir/Madam,
Thank you very much for spending a great amount of time and effort reviewing our manuscript. Your detailed comments have helped us improve the quality of our paper.
To help you navigate through the modifications, we have included a detailed response letter outlining each revision and addressing the specific points raised by you.
Yours sincerely,
Corresponding author: Ruining Jin

Reviewer 2 Report
The current manuscript entitled “Increasing supply for woody biomass-based energy through wasted resources: Insights from the US private landowners” by Vuong et al. emphasizes how utilizing the woody leftover on the land can potentially increase the woody biomass supply. After a careful reading, I found this manuscript suitable for publication in the Sustainability journal after minor revision. My specific comments are:
1. Please clearly state the objectives of this study before starting “The current study suggests …” in line 20. This sentence will better suitable after the next sentence.
2. The tense must be corrected throughout the manuscript (e.g., 101, and many others). The current version seems to be in future tense and copied from the actual research proposal. Please check this issue carefully.
3. References are must for the statement in lines 108-109.
4. Line 197 and 300 (please add date accessed).
5. Figure 1-2: please put text outside the overflowing circles.
6. Since the discussion part of the manuscript is lengthy, it would be better to provide a separate conclusion section (200 words) outlining the overall findings of this study, its limitations, and future needs.
7. References are not adjusted as per MDPI style.
Please check the accuracy of tenses and correct overall typo and grammatical issues.
Author Response

(The authors gave the same response as above.)

Reviewer 3 Report
In this article, authors presented an analysis/survey to increase the woody biomass-based energy through wasted resources which is an interesting topic. The authors have done very good research and presented a comprehensive method that shows how to increase the use of wasted woody biomass for energy. I think the manuscript can be accepted with revision in language, and some suggestions are given below.
1. Abstract needs to be modified, which must be precise and to the point. In the present form, the abstract is a little confusing. I would suggest please adding key finding clearly with quantitative results (in one or a maximum of two short sentences).
2. For understanding and increasing paper readability, I would suggest adding a flowchart that systematically presents all steps would be better.
3. Introduction section, I recommend further strengthening the introduction section by clearly describing the research gap. Please highlight the novelty and contribution of this study.
4. In section 2.1 Theoretical foundation and assumptions. Please cut this section short (cut unnecessary parts) as too many words are written, I would suggest adding the equations instead of words or for assumptions.
5. Please explain the Bayesian Mindsponge Framework, and why the author prefers to use this, as numerous authors use this framework, moreover what is the difference this paper brings with respect to the already published papers?
6. Some figures are not clear, and their quality is low, such as Figures S2 and S3. Moreover, the captions of all figures are not enough. Please give some details when writing a figure. For example. Figure X. Monthly average available wood feedstock for power generation.
7. Conclusion section is missing, I noticed that the author presents the conclusion of this paper in the discussion section. However, I strongly suggest, please add a conclusion section and copy the concluding remarks from the discussion section and place them in the conclusion section.
8. Please also highlight the limitations of this work as well as the recommendations.
Moderate improvement in writing (English language) is required.
Author Response

(The authors gave the same response as above.)

Round 2
Reviewer 1 Report
the paper was revised according to the suggestions
the paper requires minor revisions